# Accuracy of Serological Screening for the Diagnosis of Celiac Disease in Type 1 Diabetes Children

**DOI:** 10.3390/medicina59071321

**Published:** 2023-07-18

**Authors:** Chloé Girard, Aurélie De Percin, Carole Morin, Maeva Talvard, Françoise Fortenfant, Nicolas Congy-Jolivet, Claire Le Tallec, Jean-Pierre Olives, Emmanuel Mas

**Affiliations:** 1Service de Gastroentérologie, Hépatologie, Nutrition, Diabétologie et Maladies Héréditaires du Métabolisme, Hôpital des Enfants, CHU de Toulouse, 31059 Toulouse, France; girard.c@chu-toulouse.fr (C.G.); depercin.a@chu-toulouse.fr (A.D.P.); morin.c@chu-toulouse.fr (C.M.); talvard.m@chu-toulouse.fr (M.T.); letallec.c@chu-toulouse.fr (C.L.T.); jean.pierre.olives@free.fr (J.-P.O.); 2Department of Immunology, Rangueil Hospital, 31400 Toulouse, France; fortenfant.f@chu-toulouse.fr (F.F.); congy.n@chu-toulouse.fr (N.C.-J.); 3Molecular Immunogenetics Laboratory, EA 3034, Faculty of Medicine Purpan, IFR150 (INSERM), 31400 Toulouse, France; 4Faculté de Médecine, Université de Toulouse III, UPS, 31400 Toulouse, France; 5Institut de Recherche en Santé Digestive (IRSD), Université de Toulouse, INSERM, INRAE, ENVT, UPS, 31300 Toulouse, France

**Keywords:** diabetes mellitus, type 1, celiac disease, HLA antigens, antibodies, children

## Abstract

*Background and Objectives*: Patients with type 1 diabetes (T1D) are considered at high-risk for developing celiac disease (CD). The purpose of our study was to determine the prevalence of CD among children who were followed in our unit for T1D using the latest ESPGHAN guidelines, and avoiding intestinal biopsies in some of the children. Materials and *Methods*: We performed a prospective monocentric study, which included 663 T1D children between June 2014 and June 2016. We considered CD according to serological (tissue transglutaminase (TGAs) and endomysium antibodies) results. Children were included either at the time of T1D diagnosis or during their follow up. We looked for clinical and biochemical signs of CD, and for T1D characteristics. *Results*: The children’s ages ranged from 11 months to 18 years. CD was confirmed in 32 out of 663 patients with T1D, with a prevalence of 4.8%. CD was excluded in 619 children and remained uncertain for 12 children, who had positive TGAs without the required criteria. We found that 95% of T1D children express HLA-DQ2 and/or -DQ8, which was 2.4 times higher than in the general population. *Conclusions*: An intestinal biopsy could be avoided to confirm CD in the majority of T1D children. Silent forms of CD are frequent and screening is recommended for all patients. Importantly, repeated TGA assessment is required in HLA genetically predisposed T1D patients, while it is unnecessary in the 5% who are HLA-DQ2 and -DQ8 negative.

## 1. Introduction

Celiac disease (CD) is an immune-mediated systemic disorder that occurs in genetically predisposed individuals after gluten consumption. The pathophysiology of the disease is well elicited by the direct sensitization of the small intestine to gluten, or related prolamines, causing villous atrophy and resulting in various clinical presentations. Prevalence of CD is estimated to be between 0.5% and 2% worldwide, but varies considerably with geographic origin and ethnic background [1]. The European Society for Pædiatric Gastroenterology, Hepatology and Nutrition changed its guidelines for CD diagnosis in 2012 [2]. In the previous guidelines, small-bowel biopsies were required to confirm CD diagnosis [3]. Indeed, CD diagnosis was based on a pathological examination and Marsh classification to assess the degree of villous atrophy and intra-epithelial lymphocytosis. Obviously, it required an upper gastrointestinal endoscopy with duodenal or jejunal biopsies performed under sedation or general anesthesia.

Nowadays, the serological diagnosis of CD has been developed, which targets tissue transglutaminase type 2 antibodies (TGAs) and endomysium antibodies (EMAs). These antibodies were discovered in 1997 and 1983, respectively [4]. Both of them have a high sensibility and specificity. However, the techniques used are different, and are more reliable for TGAs, which is assessed by ELISA and is available in all laboratories, than for EMAs, which is detected by immunofluorescence and not available in all laboratories; moreover, EMAs may be subject to interobserver variability [5]. In addition, patients who developed CD had a genetic predisposition in genes coding for the major histocompatibility complex. Patients with Human Leucocyte Antigen (HLA) class II DQ2 and/or DQ8 are susceptible to develop CD, while DQ2/DQ8 negative patients will not [6]. HLA screening for CD in the general population is not recommended because of the high frequency of the DQ2 haplotype, more than 40%, compared to the lower prevalence of CD, about 1%. Taking into account serological and HLA typing for CD, ESPGHAN defined two different algorithms for CD diagnosis, depending on whether patients were symptomatic or if they were asymptomatic but belonged to a high-risk group [2]. Type 1 diabetes (T1D), Down’s syndrome, autoimmune thyroid disease, Turner syndrome, Williams’s syndrome, immunoglobulin A (IgA) deficiency, autoimmune liver disease and first-degree relatives with CD have a greater risk to develop CD. According to ESPGHAN’s updated guidelines, HLA genotyping is no longer necessary [7]. Indeed, HLA typing did not seem to be necessary because it was always positive for DQ2/DQ8 when EMAs were positive [8]. However, it could be useful in at-risk groups, while CD is unlikely if patients are HLA-DQ2 and HLA-DQ8 negative. The strategy for CD diagnosis in T1D children is described in Figure 1. Of note, the North American Society for Pediatric Gastroenterology and Nutrition (NASPGHAN) recommends a CD screening for T1D children, using TGA; when TGA is <3 × ULN, it should be completed by EMAs because transient mild elevation of TGA can occur in T1D [9]. Thus, an intestinal biopsy should only be performed for EMA-positive children. To note, the authors were not able to give an exact cut-off of TGAs in order to recommend intestinal biopsy.

T1D is a frequent disease with an increasing incidence in recent years. In Europe, the prevalence of T1D in children < 15 years was 94,000 in 2005, and predicted to reach 160,000 in 2020, i.e., a 1.7-fold increase [10]. T1D can be associated with other autoimmune diseases [11]. Autoimmune thyroid disease has the highest prevalence, occurring in 3–8% of young people with T1D, followed by CD [12]. The first case of patients with both CD and T1D was reported in 1969 [13]. The prevalence of biopsy-proven CD in T1D fluctuates across the world, ranging from 2.4% in Finland to 16.4% in Algeria [14]. In a recent review, the weighted pooled prevalence of CD was 5.1% [15]. However, these studies did not determine CD in T1D children according to ESPGHAN guidelines. CD screening is recommended for all T1D children for several reasons; they have a high prevalence of CD, and they are often asymptomatic (silent CD), with a risk of complications and co-morbidities like growth failure or failure to thrive, osteoporosis, retinopathy, cardiovascular complications, and intestinal cancers. A study reported that all asymptomatic T1D children with TGAs > 10 × ULN had a biopsy-proven CD [16]. The aim of our study was to determine the prevalence of CD in a large cohort of French T1D children and adolescents using HLA typing and serological markers.

## 2. Patients and Methods

### 2.1. Patients

T1D children < 18 years of age were recruited at the pediatric diabetes unit of the Children’s Hospital, Toulouse University Hospital, France, between June 2014 and June 2016. Data were prospectively collected during follow-up visits with physicians or during hospitalization at the time of T1D diagnosis.

According to 2012 ESPGHAN criteria, available at the time of the study, we looked for clinical and biological signs of CD. Typical symptoms were chronic abdominal pain, chronic or intermittent diarrhea, chronic constipation, cramping or distension, vomiting, failure to thrive, weight loss, stunted growth, and iron-deficiency anemia. Atypical symptoms were delayed puberty, chronic fatigue, anorexia, irritability and attention disorder, and abnormal liver function tests. Patients were considered symptomatic if at least one criterion was present.

We also collected other data: breastfeeding in infancy, autoimmune diseases in relatives, age at onset of T1D, specific antibodies of T1D (islet antibodies, glutamic, and dicarboxylase antibodies), weight, and height at inclusion. Blood samples were also collected for hemoglobin, ferritin, iron, liver enzymes, hemoglobin A1c (HbA1c), TGAs, EMAs, and HLA typing that was required between 2012 and 2020. All these tests were performed at the laboratory of Toulouse University Hospital.

We assessed the metabolic control of T1D by using HbA1c levels. According to international guidelines for children and teenagers with T1D, we defined a good control of diabetes as an average of HbA1c measures during the last year of <7.5% [17]. This criterion was not used for patients who were included at the onset of T1D.

When CD diagnosis was performed prior to June 2014, these patients were not screened but we retrospectively looked for the main data in their medical records and with a questionnaire filled in by the parents. We took them into account for CD prevalence.

### 2.2. Detection of Auto-Antibodies

TGA were measured using Luminex methodology with Bioplex 2200 Celiac IgA and IgG kits (Bio-Rad Laboratories Inc., Hercules, CA, USA). When patients had an IgA deficiency, TGA IgGs were automatically measured [18]. Positive TGA were defined as titers > ULN (15 UI/mL). Thereafter, we separated them into three groups: TGAs ≤ 3 × ULN, 3 < TGAs < 10 × ULN, and TGAs ≥ 10 × ULN.

For EMAs, the laboratory used staining of rhesus monkey esophagus substrate by Indirect Immunofluorescence Assay, and results were positive or negative (kit NOVA Lite Ensomysial, Inova Diagnostics Inc., San Diego, CA, USA).

When T1D was diagnosed in our unit, islet antibodies and/or glutamic and dicarboxylase antibodies were performed to confirm autoimmunity. We collected these results in patient files.

### 2.3. HLA Typing

Written informed consent of parents was obtained for HLA typing. Blood sample was drawn and genomic DNA analysis of HLA susceptibility for T1D and CD was performed on Luminex 200 using LABType SSO One lambda^®^ (Thermo Fisher Scientific, Meerbusch, Germany). Genetically, CD susceptibility was defined by the presence of DQ2 and/or DQ8 expression.

### 2.4. Duodenal Biopsies

When a pathological confirmation was required, a pediatric gastroenterologist performed an upper gastrointestinal endoscopy. A pathologist reviewed biopsies and determined CD according to Marsh’s criteria. Briefly, CD was confirmed if pathological findings were Marsh 3 or 4 (total or partial villous atrophy, crypt hyperplasia, increased intra-epithelial lymphocytes) [19].

### 2.5. CD Determination

Children were considered having CD if they had TGAs ≥ 10 × ULN or if they were symptomatic with 3 < TGAs < 10 × ULN and biopsy-proven CD (Figure 1). To note, other patients with biopsy-proven CD were also taken into consideration.

We separated the remaining children in the uncertain group (asymptomatic children with TGAs < 10 × ULN) and the non-celiac group (TGAs < 1 × ULN).

### 2.6. Ethical Consideration

The Ethics Committee of Toulouse University Hospital approved the study protocol (n°68-0914). Verbal and written information about the study were given to the patients and their parents, and written informed consent was obtained.

### 2.7. Data Analysis

Numerical data were expressed as means ± standard error of the mean. We performed a descriptive analysis of the main clinical and biological characteristics of the patients. Statistical analysis was performed using unpaired Student *t* test and χ^2^ test for continuous or categorical variables, respectively. Values of *p* < 0.05 were considered significant.

## 3. Results

We included 663 T1D children (331 boys and 332 girls), and 32 of them had CD (Figure 2). CD was excluded in 619 children and 12 remained uncertain for CD because they had a positive TGAs without the required criteria. CD prevalence was 4.8% (Table 1).

CD screening was performed at the onset of T1D in 51 children, while 612 were included during their follow-up. The mean age at the time of T1D diagnosis was 6.6 ± 3.8 years. At the time of inclusion, children were 11.1 ± 4.2-year-old (0.9–18), and T1D duration was 4.5 ± 3.9 years (0–17.1). The proportion of children that were breastfed was 53.7%. Specific T1D antibodies were found in 89.4% of the 527 patients tested for islet antibodies and/or glutamic and dicarboxylase antibodies.

The age at CD diagnosis was 10.4 ± 4.9 years (1.7–17.9). A CD diagnosis preceded T1D in one patient only, by 3 years. The diagnosis of CD was performed at the time of T1D diagnosis in nine children. CD diagnosis was performed 6.1 ± 4.5 years (maximum of 15 years) after T1D onset in 22 children. Finally, the mean delay between diagnosis of T1D and CD was 4.1 ± 4.8 years.

The age at T1D diagnosis did not differ between CD and non-CD patients (Table 1). At the time of study inclusion, children in the uncertain group were younger than children in the CD group (*p* < 0.05) and non-CD group (*p* < 0.01).

A family history of autoimmune diseases was present in 53% (n = 17) of CD patients, and in 45% of non-CD children (*p* < 0.0001). We found a personal history of autoimmune thyroid disease in 40 out of 663 patients (6%), including three children with associated CD.

Among symptoms, which were present in 38.9% of the children, the most frequent were chronic abdominal pain (17.4%), chronic constipation (12.4%), and irritability and attention disorders (9.1%). In the CD group, 23 patients (72%) had symptoms (Figure 3A), and the clinical manifestations were variably combined and consisted of gastrointestinal symptoms (in 16 patients, including abdominal distention, constipation, vomiting, diarrhea, and/or abdominal pain), failure to thrive (in 5 patients), or atypical symptoms (in 7 patients), including delayed puberty, chronic fatigue, anorexia, irritability, and attention disorder.

Iron deficiency anemia was found in four patients with CD; it was the only sign for three of them who had no clinical symptoms. Liver enzymes were within the normal range in all CD patients. HbA1c levels were significantly higher for CD patients than non-CD patients (*p* = 0.015) (Figure 3B). There was no other statistically significant difference between these groups for biological parameters (Table 1).

HLA typing was performed in 645 patients and 95% of them had genetic susceptibility for CD with HLA-DQ2 and/or -DQ8 positivity; no patients who were HLA-DQ2 and -DQ8 negative had CD (Figure 4A,B). Among celiac children, 86% were DQ2-positive, 52% were DQ8-positive, and 38% were DQ2- and DQ8-positive (Figure 4B). Among non-celiac children, 75%, 57%, and 37% were DQ2-, DQ8-, DQ2- and DQ8-positive, respectively (Figure 4B). All of the children of the uncertain group were DQ2-(100%) and DQ8-positive (75%).

We found that 12 patients of 663 (1.8%) had an IgA deficiency. Only one patient with IgA deficiency had CD: IgG TGAs > 10 × ULN, positive IgG EMAs, DQ2 haplotype, and typical symptoms of CD. CD patients can be displayed according to TGAs levels: 28 had TGAs ≥ 10 × ULN, 1 had TGAs between 3 to 10 × ULN and 3 had TGAs ≤ 3 × ULN. All patients with CD had positive EMAs, except for one who was diagnosed before the beginning of the study and the EMA result was not available. CD was diagnosed before the study in seven children: three of them underwent intestinal biopsies that found villous atrophy and others had all criteria for non-invasive diagnosis. At the time of the study, 25 children were diagnosed; 22 had TGAs ≥ 10 × ULN and 3 other children had TGAs ≤ 3 × ULN and biopsies, which were performed because they were symptomatic or had a familial history that confirmed CD. Overall, 12 patients underwent biopsies that confirmed CD in 10 children. Biopsies ruled out CD in two children who were symptomatic and had persistent positive TGAs < 3 × ULN.

## 4. Discussion

Using non-invasive markers of CD in the majority of cases, we found a 4.8% prevalence of CD in this large monocentric cohort of French T1D children. This prevalence was similar to the result of a meta-analysis (5.1%) [15] and with studies performed in Italy [20], the UK [21], Israel [22], and Serbia [23]. Meanwhile, the prevalence of biopsy-proven CD in T1D fluctuates across the world, ranging from 2.4% in Finland to 16.4% in Algeria [14]. Interestingly, our results were in agreement with the 4.4% prevalence of biopsy-proven CD in T1D found in a large Turkish study [24]. The variability of biopsy-proven CD prevalence could be explained by a bad orientation of the histological slides [25] and by the transient increase in low TGA levels in T1D [26]. This high prevalence of CD in T1D children can be explained by HLA differences across the world. In our study, we found that 95% of T1D were positive for HLA-DQ2/DQ8. This result is nearly identical to the HLA typing performed in 176 T1D Scottish children (94%) [27], and in 121 TID Austrian children (92%) [28].

The screening and diagnosis of CD in T1D children are challenging. According to the guidelines of the American Diabetes Association (ADA) [29], the International Society for Pediatric and Adolescent Diabetes (ISPAD) [12], and the NASPGHAN [9,30], T1D children should be screened by TGA and/or EMA at the time of diagnosis and at different time points thereafter. With persistent CD antibodies, children should be referred to intestinal biopsy. Since 2012, the ESPGHAN guidelines recommended not to perform an intestinal biopsy in symptomatic children with high levels of TGAs (>10 × ULN) who also had positive EMAs [2,7]. A prospective study confirmed that symptomatic children with TGAs > 10 × ULN and positive EMAs had a correct CD diagnosis with omission of biopsies [8]. There was a lack of adherence to ESPGHAN guidelines, and only 14% of respondents to questionnaires used HLA typing for the first-line testing [21]. Moreover, some differences exist between ESPGHAN and NASPGHAN recommendations [2,7,9]. We believe that a consensus for CD screening and diagnosis for this particular high-risk group is mandatory. In our study, we found that HLA genotyping could only definitively exclude CD in 5% of T1D children. Thus, the cost-effectiveness of HLA testing (around EUR 190) is under debate [27,31]. Serological markers of CD were less expensive and useful, but they had to be reassessed frequently in order to follow TGA titers. Castellaneta et al. found that among 446 T1D children, 65 had increased TGAs; all children with persistently elevated TGAs had villous atrophy while 2/3 of children with low TGAs titers became negative [32]. Another study revealed a rate of TGA normalization of 35.4% [26]. These normalizations occurred while children were always consuming gluten. TGA normalization was increased in children with low TGA titers [26]. In the literature, villous atrophy was reported in 50–60% and 44–100% of asymptomatic T1D with positive TGAs or EMAs respectively [33]. We found that 6.8% of our T1D children were positive to TGAs. To note, we found that 1.8% of our children had an IgA deficiency. The prevalence of IgA deficiency is 10 times higher in CD than in the general population [33]. In the Swedish registry, CD was found in 6.7% of patients with IgA deficiency and in 0.19% of controls; T1D was found in 5.9% and 0.57% [34]. IgA deficiency is lower in our cohort, and we did not find any difference for IgA deficiency between CD and non-CD T1D children. Nevertheless, serological CD screening remains effective while TGA IgGs were positive in children with IgA deficiency. This study was designed to perform duodenal biopsy to look for villous atrophy and intra-epithelial lymphocytes in symptomatic children whose TGA levels were between 3 and 10. Even if a duodenal biopsy was not required for other children, three children with TGA levels ≤ 3 × ULN and positive EMAs underwent endoscopy because of symptoms or familial history of CD; all of them had biopsy-proven CD. Four children with TGAs ≥ 10 × ULN underwent duodenal-biopsy that confirmed CD in all of them. Finally, EMAs were only negative in four children with low levels of TGAs (<3.1 × ULN), maybe because of a transient mild elevation of TGA as it was already described [9].

Clinical and biological symptoms of CD do not seem to be relevant in T1D children. Indeed, CD is generally diagnosed during its silent or potential phases in T1D children, i.e., with or without histological abnormalities of the small intestine, respectively [33]. In our study, we did not find significant differences between CD and non-CD T1D children for intestinal symptoms, failure to thrive, growth retardation, iron deficiency, and anemia, for example. Kakleas et al. found that more than 50% of children with T1D and positive TGAs had no or very mild symptoms of CD [35]. A study compared T1D patients with CD detected by serological screening (n = 22) or by clinical suspicion (n = 498) [36]. They found that serological screening prevented a decrease in growth; moreover, the adherence and the response to a gluten-free diet were similar. In the literature, CD remains silent for a long time in T1D patients and these children had normal growth. However, in Germany and Austria, a multicenter survey revealed that biopsy-proven CD (0.6% in 1995 and 1.3% in 2008) had a significantly lower weight and height standard deviation score that persisted after a 5-year follow-up [37]. T1D with CD should have a milder phenotype than CD alone; 41 biopsy-proven CD children with T1D had significantly higher height, weight and body mass index (BMI), lower anemia, and better bone health than children with CD alone after a 3-year follow-up [38]. T1D children with CD were less frequently symptomatic than children with CD alone, 48% vs. 75%, respectively [38]. In another study of T1D children, the authors reported that only 26% of biopsy-proven CD were symptomatic and that CD did not significantly alter weight, height, BMI, and HbA1c [22]. Altogether, these data support an active screening of CD in T1D children.

CD generally appears a few years after the onset of T1D. In a meta-analysis, CD was diagnosed prior to T1D in 7% [15]. When CD appeared after T1D onset, it occurred within the first year in 40%, within 2 years in 55%, and within 5 years in 79% [15]. Other risk factors to develop CD in T1D children are gender and age at T1D diagnosis. It was reported that female gender could increase the risk of CD in this population. Such as in other studies [22,32], we did not find statistically significant differences between CD and non-CD children in our study for gender. CD children were younger than non-CD children at the time of T1D diagnosis [20,32,39]. CD incidence was 10.4 vs. 6.4 per 1000 person-years in children aged < 5 or ≥5 years at T1D diagnosis, respectively [39]. Moreover, CD diagnosis was performed after 2, 5 and 10 years of T1D onset in 45, 78 and 94% of cases, respectively [39]. However, other studies did not find significant differences for age at onset of T1D in children with or without associated CD [22]. If CD symptoms are not useful to the diagnosis of CD in T1D children, HbA1c should be taken into account. We found that T1D children with CD had a higher level of HbA1c during the year prior to the study screening. However, other studies did not find that metabolic control of T1D (HbA1c, episodes of hypoglycemia or ketoacidosis) was statistically different according to the presence or absence of CD [22].

It is important to make an accurate CD diagnosis in T1D in order to start a gluten-free diet (GFD) when necessary. Depending on ADA, ISPAD or NASPGHAN, a GFD should be started in symptomatic or even in asymptomatic T1D children with CD [9,12,29,30]. It is well known that a GFD can have psychological consequences, but T1D patients with CD have an increased risk of diabetic retinopathy, cardiovascular complications, and a reduced life expectancy in comparison to non-CD T1D patients. The compliance to a GFD is lower in T1D with CD than in patients with CD alone [40]. The additional diagnosis of CD could have minimal impact on quality of life in children with T1D, but compliance with a GFD is an essential factor to obtain an optimal quality of life [41,42]. A case-control study including 35 youths with T1D alone and 35 youths with T1D and CD confirmed the importance of GFD compliance [43]. The results of generic and diabetes-specific quality of life assessment were similar between both groups. However, the results were better in adherent vs. non-adherent children to a GFD. As expected, glycemic control was improved by the adherence to a GFD. In CD patients who do not adhere to a GFD, the risk of intestinal malignancies increases. CD affected the risk of diabetic retinopathy in T1D patients after 10 years of follow-ups [44]. Mollazadegan et al. also reported that a CD course of ≥15 years was associated with a 2.8-fold increased risk of death in T1D patients [45]. When asymptomatic patients had CD, a study showed that gluten restriction increased bone mineralization [46]. The results of an ongoing randomized multicenter trial, which aims to assess the efficacy of GFD in asymptomatic CD children with T1D, should help us to determine the indications for dietary interventions [47].

Our screening protocol revealed a prevalence that seems in agreement with recent studies [20,23], but it should enable us to reduce the number of endoscopies. Indeed, T1D children with high levels of TGA had biopsy-proven CD even if they are asymptomatic [23]. To note, a study found 100% sensitivity and 99% specificity for the prediction of CD when TGA titers were >3 × ULN [48]. Another point to discuss is whether or not to perform HLA screening of CD in T1D. Even if this test is more expensive than serological CD screening, we believe that it could be performed at the time of T1D diagnosis and be relevant for 5% of these children. We do not agree with Binder et al. who do not recommend HLA screening in T1D patients because it should be positive for the majority of them [28]. Indeed, on the one hand, repeated TGAs and/or EMAs could be avoided during their follow-up, improving the cost-effectiveness of HLA genotyping. On the other hand, these children could have psychosocial benefits because they should already know that they were not at risk of developing CD later on. Therefore, active screening of CD in T1D children should include HLA screening at diagnosis and monitoring of TGAs. EMAs should be performed at least for children with TGAs > 3 × ULN. EMAs seem to have a good correlation with intestinal villous atrophy and their positivity should be sufficient to confirm CD in the high-risk group of T1D children.

## 5. Conclusions

In order to confirm CD, an intestinal biopsy could be avoided in the majority of T1D children. Serological markers are useful and reliable for CD screening even in the setting of this high-risk group of patients. As most of the T1D children have a silent form of CD, symptoms are not reliable. Other studies should determine a better follow-up protocol for uncertain CD in T1D.

## Figures and Tables

**Figure 1 medicina-59-01321-f001:**
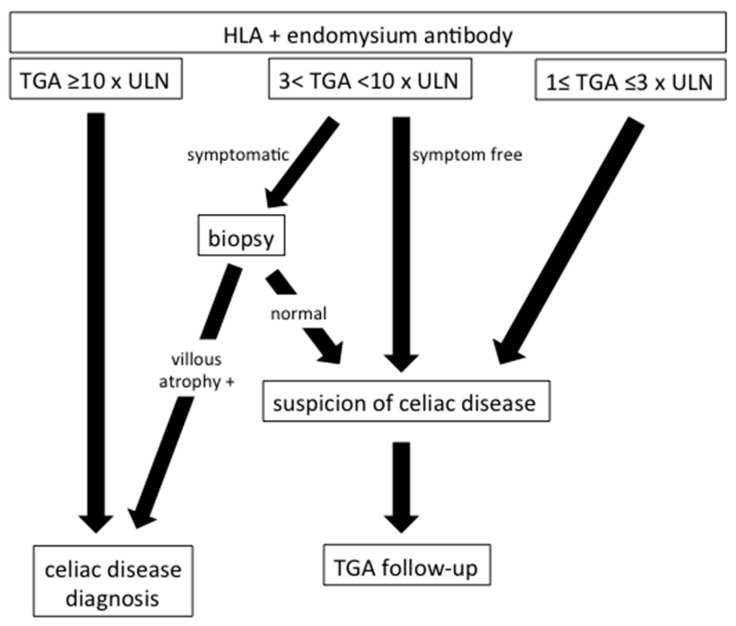
Celiac disease’s assessment in type 1 diabetes children. Abbreviations: HLA, human leucocyte antigen; TGA, tissue transglutaminase type 2 antibodies; and ULN, upper limit of normal.

**Figure 2 medicina-59-01321-f002:**
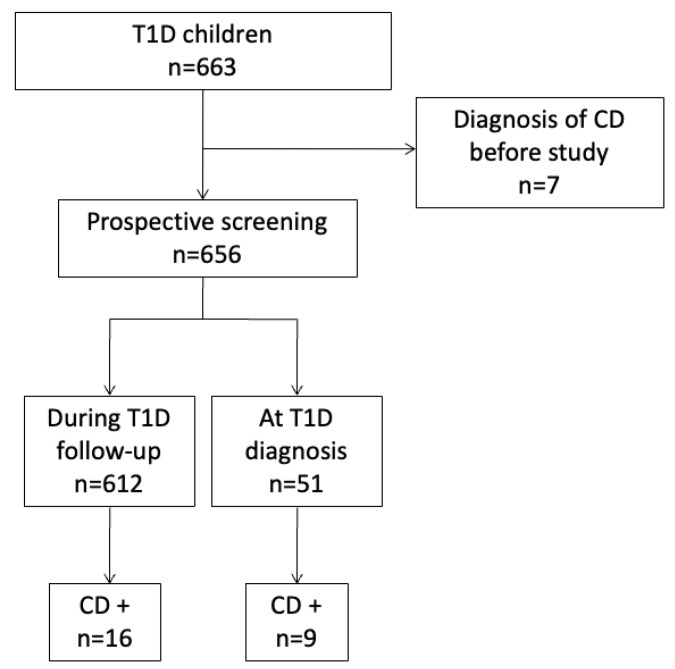
Flow chart of the study population. Abbreviations: CD, celiac disease; T1D, type 1 diabetes.

**Figure 3 medicina-59-01321-f003:**
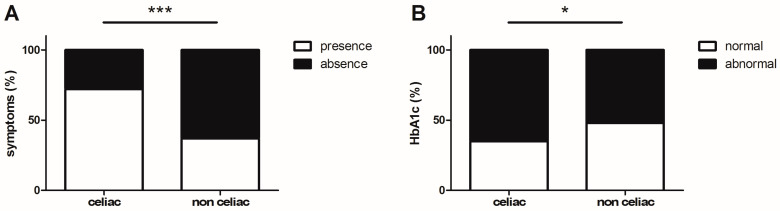
Symptoms compatible with celiac disease (**A**) and control of type 1 diabetes defined by an HbA1c cut-off of 7.5% (**B**). * *p* < 0.05 and *** *p* < 0.001.

**Figure 4 medicina-59-01321-f004:**
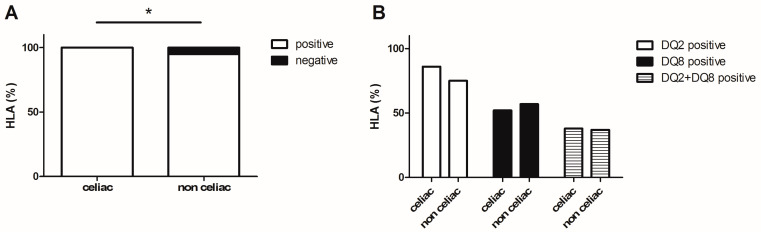
HLA typing. Children at-risk of celiac disease who were HLA-DQ2 or -DQ8 positive (**A**) and results according to HLA-DQ2, -DQ8 and -DQ2/DQ8 positivity (**B**). * *p* < 0.05.

**Table 1 medicina-59-01321-t001:** Clinical and biological characteristics of children.

	Celiac (n = 32)	Uncertain (n = 12)	Non-Celiac (n = 619)
Sex ratio	0.6	0.3	0.5
Age at diabetes diagnosis (years)	6.2 ± 0.7	5.7 ± 1.1	6.6 ± 0.2
Age at study inclusion (years)	11.3 ± 0.8 ^¤¤^	7.5 ± 0.9 ^$$^	11.2 ± 0.2
Weight (SD)	0.74 ± 0.25	0.73 ± 0.45	0.98 ± 0.06
Height (SD)	0.54 ± 0.24	0.84 ± 0.38	0.74 ± 0.05
HbA1c (%)	7.9 ± 0.2 *^, ¤^	7.1 ± 0.2	7.5 ± 0.4
BMI (z-score)	0.35 ± 0.23	0.10 ± 0.42	0.51 ± 0.05
Hb (g/L)	1.35 ± 0.02	1.38 ± 0.03	1.35 ± 0.01
Iron (µmol/L)	14.8 ± 1.4	14.3 ± 0.8	14.7 ± 0.3
Ferritin (µg/L)	72.3 ± 16.9	58.7 ± 12.1	66.9 ± 2.3
ASAT (IU/L)	25.9 ± 2.0	26.8 ± 2.1	24.7 ± 0.3
ALAT (IU/L)	21.3 ± 1.4	18.9 ± 2.4	18.7 ± 0.3

Abbreviations: ALAT, alanine aminotransferase; ASAT, aspartate aminotransferase; BMI, body mass index; Hb, hemoglobin; HbA1c, hemoglobin A1c; SD, standard deviation; * celiac vs. non-celiac, ^¤^ celiac vs. uncertain, ^$^ uncertain vs. non-celiac; *, ^¤^
*p* < 0.05 and ^¤¤^, ^$$^
*p* < 0.01.

## Data Availability

The data presented in this study are available on request from the corresponding author.

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
