# Peer review of "Accuracy of Serological Screening for the Diagnosis of Celiac Disease in Type 1 Diabetes Children"

_medicina, 2023, doi:10.3390/medicina59071321_

Round 1

Reviewer 1 Report

Dear Editor,

I should first thank for inviting me as potential reviewer to read and comment on paper entitled ‘’Accuracy of serological screening for the diagnosis of celiac 2 disease in type 1 diabetes children’’.

In the current study, the authors aimed to determine the prevalence of CD in a large cohort of French T1D children and adolescents using HLA typing and serological markers..

The main title accurately reflects the major topic and content of the study.

The abstract summarizes and reflects the work described in the manuscript. Also, the abstract presents the significant points related to the background, objectives, materials and methods, results and conclusions.

The materials and methods sufficiently described for the results and conclusions that are presented in the preceding sections. The study type and design were defined in the section of the materials and methods. Figures and tables are sufficient and well-organized. Ethics Committee approval was received. So, the section materials and methods is adequate.

The statistical methods used are appropriate.

The section of the discussion is well organized. The conclusions are drawn appropriately supported by the literature. The manuscript adequately describes the background, present status and significance of the study. The manuscript interprets the findings adequately and appropriately, highlighting the key points clearly.

I think that it will contribute to the literature. I have some minor crticisms.

-       According to ESPGHAN updated guidelines, HLA genotyping is no more necessary. I noticed that HLA analysis was performed in the study. I think because it was a retrospective study, analyzes were done at that time. Please state it in the manuscript.

-       In the current study, the authors found a 4.8% prevalence of CD in that large monocentric cohort of French T1D children. The similar result 4.4% was detected in recent study conducted in Turkey (Iran J Pediatr 2020;30:e97306). Also, the similar symptoms and results were detected in celiac patients in that study. If the recent published article acalled ‘’Prevalence of Celiac Disease in Children with Type 1 Diabetes Mellitus in the South of Turkey’’ are cited, the manuscript would be better.

Author Response

Thank you for reviewing our manuscript.

Please find our answer:

-       According to ESPGHAN updated guidelines, HLA genotyping is no more necessary. I noticed that HLA analysis was performed in the study. I think because it was a retrospective study, analyzes were done at that time. Please state it in the manuscript.

Indeed, the study was performed before the publication of the 2020 updated ESPGHAN guidelines. Thus, HLA genotyping was performed according to 2012 ESPGHAN criteria. We added this statement.

 -       In the current study, the authors found a 4.8% prevalence of CD in that large monocentric cohort of French T1D children. The similar result 4.4% was detected in recent study conducted in Turkey (Iran J Pediatr 2020;30:e97306). Also, the similar symptoms and results were detected in celiac patients in that study. If the recent published article acalled ‘’Prevalence of Celiac Disease in Children with Type 1 Diabetes Mellitus in the South of Turkey’’ are cited, the manuscript would be better.

We thank the reviewer for this comment. We added this reference in the manuscript “Interestingly, our results were in agreement with the 4.4% prevalence of biopsy-proven CD in T1D found in a large Turkish study”.

Reviewer 2 Report

Here are my comments regarding the article by Girard et al. entitled "Accuracy of serological screening for the diagnosis of celiac 2 disease in type 1 diabetes children"

- The title states that you studied the accuracy of celiac disease serology, however, I believe that this has been studied a lot and is clear that celiac antibody tests are very good. The interesting point here are the type 1 diabetes patients. Maybe change the title to something more precise like "Prevalence and serological screening for.."

- In the introduction the authors elaborate very specifically the diagnostic algorithm. This is boring and difficult to follow; this could be easily replaced with the algorithm itself (refer to figure 1?) and so the introduction would be much shorter and interesting for the reader. 

- Please elaborate data about the patients that were ruled out with biopsies. Any inflammation? EMA negative? Pictures of histological slides might be good as there is a high rate of false negative biopsy results. Follow-up on these patients available? Were biopsies well-oriented? see werkstetter pape in gastroenterology 2017 and Taavela Plos One 2013

- please specify how the genetic assessments were done, what kit was used.

Author Response

Thank you for reviewing our manuscript.

- The title states that you studied the accuracy of celiac disease serology, however, I believe that this has been studied a lot and is clear that celiac antibody tests are very good. The interesting point here are the type 1 diabetes patients. Maybe change the title to something more precise like "Prevalence and serological screening for.."

We apologize but we believe that the current title is correct, as well as reviewer 1 believed. Thus, we did not perform this change.

- In the introduction the authors elaborate very specifically the diagnostic algorithm. This is boring and difficult to follow; this could be easily replaced with the algorithm itself (refer to figure 1?) and so the introduction would be much shorter and interesting for the reader. 

This comment was taken into account and sentences were deleted.

- Please elaborate data about the patients that were ruled out with biopsies. Any inflammation? EMA negative? Pictures of histological slides might be good as there is a high rate of false negative biopsy results. Follow-up on these patients available? Were biopsies well-oriented? see werkstetter pape in gastroenterology 2017 and Taavela Plos One 2013

We believe that it is correct to discuss the importance of well-oriented biopsies in order to prevent false negative biopsy results. However, we only had 2 negative biopsy results in patients who also had very low TGA antibodies. As you know, low TGA levels could be transient in T1D. Thus, we are not sure that it would be useful to add histological sides or comments. However, we added a sentence in the discussion section and Taavela reference.

- please specify how the genetic assessments were done, what kit was used.

This information was added in the method section.

Reviewer 3 Report

This is a timely and well crafted study. Endpoints and outcomes are clearly defined and the analysis is sound. The paper is of high relevance to the discipline of Gastroenterology.

Author Response

Thank you for reviewing our manuscript and for your interest.